# Identification of *SLC15A4/PHT1* Gene Products Upregulation Marking the Intestinal Epithelial Monolayer of Ulcerative Colitis Patients

**DOI:** 10.3390/ijms232113170

**Published:** 2022-10-29

**Authors:** Aurora Mazzei, Grazia Serino, Alessandro Romano, Emanuele Piccinno, Viviana Scalavino, Anna Maria Valentini, Raffaele Armentano, Roberta Schiavone, Gianluigi Giannelli, Tiziano Verri, Amilcare Barca

**Affiliations:** 1Department of Biological and Environmental Sciences and Technologies (DeBEST), University of Salento, 73100 Lecce, Italy; aurora.mazzei@unisalento.it (A.M.); roberta.schiavone@unisalento.it (R.S.); tiziano.verri@unisalento.it (T.V.); 2National Institute of Gastroenterology “S. de Bellis”, IRCCS Research Hospital, 70013 Castellana Grotte, Italy; emanuele.piccinno@irccsdebellis.it (E.P.); viviana.scalavino@irccsdebellis.it (V.S.); am.valentini@irccsdebellis.it (A.M.V.); raffaele.armentano@irccsdebellis.it (R.A.); gianluigi.giannelli@irccsdebellis.it (G.G.); 3Division of Neuroscience, Experimental Neurology Unit, San Raffaele Scientific Institute, 20132 Milano, Italy; romano.alessandro@hsr.it

**Keywords:** SLC15A4/PHT1, inflammation, enterocytes, IBD, ulcerative colitis

## Abstract

SLC15A4/PHT1 is an endolysosome-resident carrier of oligopeptides and histidine recently come into view as a key path marker of immune/autoimmune/inflammatory pathways in immune cells. Yet, its emerging role in inflammatory processes directly targeting the gastrointestinal epithelial layer, as in the multifactorial pathophysiology of inflammatory bowel disease (IBD), is poorly investigated. Here, the first identification of *SLC15A4/PHT1* gene products in human colonic epithelium of ulcerative colitis (UC) patients is reported, showing protein primarily localized in intracellular vesicle-like compartments. Qualitative and quantitative immunohistochemical analyses of colon biopsies revealed overexpression of SLC15A4/PHT1 protein product in the epithelial layer of UC patients. Results were successfully mirrored *in vitro*, in spontaneously differentiated enterocyte-like monolayers of Caco-2 cells specifically exposed to DSS (dextran sodium sulphate) to mimic IBD inflammatory onsets. SLC15A4/PHT1 expression and cellular localization were characterized confirming its (dys)regulation traits in inflamed *vs.* healthy epithelia, strongly hinting the hypothesis of SLC15A4/PHT1 increased function associated with epithelial inflammation in IBD patients.

## 1. Introduction

Research on therapies for inflammatory diseases such as intestinal bowel disease (IBD) typically investigates cytokines and/or cell surface receptors that directly control downstream immune responses or signaling molecules. Examples are studies on antibody drugs against IL-6 and TNF-α which have successful anti-inflammatory effects in IBD patients. Nevertheless, it is established that targeting a single molecule is only partially effective and additional administration of multiple drugs directed to various targets is usually required to efficiently control the disease. In this respect, processes involving proteins of endosomes, lysosomes, and related intracellular vesicles are being increasingly investigated to find out new potential therapeutic routes for treating inflammatory diseases [1]. Endolysosomes are subcellular organelles central in regulating membrane repair, autophagy, cell death, endocytosis, and other aspects of cellular homeostasis [2,3]. Many features of endolysosomes, including internal pH, levels of amino acids, ions content and transport processes across membranes are finely tuned to optimize signaling events: dysregulation of these properties causes disturbance of their functions including signal transduction [4]. The solute carrier family 15 member A4 (SLC15A4), also known as peptide/histidine transporter 1 (PHT1), is an endolysosome-resident transporter which transports histidine, oligopeptides with certain specificity for histidine-containing dipeptides, and NOD1 ligands as L-Ala-γ-D-Glu-meso-diaminopimelic acid (tri-DAP), from inside the lysosome outwardly to the cytosol exploiting proton gradients [5,6].

*SLC15A4* gene is specifically expressed in immune cells such as B cells and plasmacytoid dendritic cells [7,8] playing a critical role in autoimmune and other inflammatory diseases: genome-wide analyses as well as data from murine models indicate that it is closely associated with processes causing type 2 diabetes [9], systemic lupus erythematosus (SLE) [10] and IBD [6,7,11]. In fact, SLC15A4/PHT1 is critical for TLR7- and TLR9-dependent production of cytokines including type I interferons (IFN-I) and IL-6 [6,12,13]. In addition, SLC15A4/PHT1 colocalizes with mechanistic targets of rapamycin (mTOR) in LAMP1+ compartments and is also important for the regulation of mTORC1 activity, that is required not only for TLR7- and TLR9-mediated signaling but for IFN-regulatory factors (IRFs) activation also [7]. SLC15A4/PHT1 mediates TLR7/TLR9-dependent signaling through several mechanisms. One mechanism involves its transporter activity [7]. As a carrier, SLC15A4/PHT1 is acknowledged for transport activity of histidine from the endolysosome into the cytosol, and thus SLC15A4/PHT1 deficiency causes histidine accumulation in the endolysosomal compartment. Histidine is the only amino acid possessing an imidazole ring and has unique acid–base properties (pKa ~6), and so it is highly effective in controlling pH; in this respect, the accumulation of histidine in endolysosomes prevents the vesicular acidification by buffering action [7]. De-acidification of endolysosomes interferes with activation of the mTORC1 signaling axis; hence, SLC15A4/PHT1 is crucial to control endolysosomes pH in TLR7/TLR9 and IFN-I circuits [7]. Another mechanism through which SLC15A4/PHT1 mediates TLR7/TLR9-dependent signaling is its scaffolding function with respect to IRF5 [14]. This function is exerted by association with the adaptor protein TASL (“TLR adaptor interacting with SLC15A4 on the lysosome”), encoded by the SLE-associated gene, CXorf21 [15,16]. Deletion of SLC15A4/PHT1 or TASL specifically impairs the activation of the IRF pathway without affecting NF-κB and MAPK signaling, that indicates that ligand recognition and TLR engagement in the endolysosome occur normally. Furthermore, TASL contains a conserved motif (namely pLxIS) that mediates the recruitment and activation of IRF5. This finding shows that TASL is an innate immune mediator for TLR7, TLR8, and TLR9 signaling, revealing a clear mechanistic analogy with the IRF3 adaptors STING, MAVS, and TRIF. The identification of TASL as a component linking endolysosomal TLRs to the IRF5 transcription factor via SLC15A4/PHT1 provides a mechanistic explanation for the involvement of these proteins in inflammatory disease processes. Finally, a novel function of SLC15A4/PHT1 has been further revealed in the maintenance of mitochondrial integrity required to maintain autophagy [11].

In mouse models, the loss of *SLC15A4* gene products significantly ameliorates the symptoms of several diseases, presumably by affecting multiple signaling pathways mediated by TLR7/TLR9, IRFs, and mTORC1. For example, in mouse models of SLE, loss of SLC15A4/PHT1 suppresses the IFN-I secretion and production of pathogenic anti-dsDNA and anti-snRNP autoantibodies. Katewa et al. also demonstrated that lack of SLC15A4/PHT1 protects mice from developing spontaneous lupus nephritis and from lupus disease development after IFN-α administration [10]. In an acute psoriasiform dermatitis model, SCL15A4 *feeble* mice (mice with one base-pair mutation in the SCL15A4 gene that renders the transport activity inactive) are resistant to TLR-7 agonist imiquimod inducing weight loss and epidermal skin thickening [16]. Finally, SLC15A4/PHT1 protein is also a potential therapeutic target in patients with colitis. *SLC15A4/PHT1*-deficient mice show alleviation of DSS-induced colitis [6]. In mucosal immune cells, as dendritic cells and macrophages, SLC15A4 presumably promotes gut inflammation by mediating NOD1- and TLR9-dependent inflammatory responses [6]; therefore, it is conceivable that SLC15A4/PHT1 loss efficiently ameliorates gut inflammation by diminishing both TLR9-mediated and NOD1-mediated signaling.

To date, the role of the *SLC15A4/PHT1* gene in cellular physiology is not fully characterized and the functional importance of this transporter is being more deeply investigated only recently. The major information presented to date in the literature highlight the evidence that the SLC15A4/PHT1 protein pivotally and widely affects inflammatory and metabolic signaling at the endolysosome level, and that functional loss of it inhibits multiple inflammatory signals in various inflammatory cellular settings. Several studies describe the association of SLC15A4/PHT1 “gain-of-function” with the immune/autoimmune/inflammatory processes, also concerning GI (gastrointestinal) inflammatory diseases; nevertheless, the relevant knowledge is mainly limited to immune cells, still lacking established information regarding intestinal cells and tissue districts. In this work, we aimed to characterize the physiological and pathophysiological expression features of the *SLC15A4/PHT1* gene products in the intestinal epithelial monolayer in the context of responses to stimuli and processes eliciting inflammation of the GI epithelial barrier, and to identify SLC15A4/PHT1 expression and functional significance in the context of inflammatory alterations related to the human IBD etiological framework.

## 2. Results

### 2.1. DSS Effects on Inflammation Genes Expression in Caco-2 Monolayers

The transcriptional expression of *IL-1β*, *IL-6*, *NF-kB1,* and *IRF5* inflammation genes and *CASP3* (apoptosis) were determined using qPCR in Caco-2 monolayers after DSS administration. Analysis of IL-1β mRNA showed that DSS exposure determined a significant dose-dependent downregulation compared to untreated control monolayers (*p* < 0.001) (Figure 1). On the other hand, DSS treatments resulted in a significant dose-dependent increase in the expression of IL-6 and NF-kB1 mRNAs compared to untreated cells. CASP3 mRNA was found significantly increased by treatments in a DSS dose-dependent manner. IRF5 mRNA levels were not significantly affected by DSS treatments.

### 2.2. SLC15A4/PHT1 mRNA Expression Variations Triggered by DSS Challenge

Levels of SLC15A4/PHT1 mRNA were evaluated in mature Caco-2 monolayers challenged with 2% or 4% DSS (Figure 2). qPCR analysis revealed that mRNA levels showed an overall DSS dose-dependent up-regulation trend compared to untreated cells (ctrl = 100%), which became statistically significant following DSS 4% (156.6% ± 13.0; *p* < 0.05).

### 2.3. DSS Proinflammatory Trigger Affects SLC15A4/PHT1 Intracellular Localization in Enterocyte-Like Caco-2 Monolayer

The cell-specific localization of the SLC15A4/PHT1 protein was established in mature Caco-2 monolayers differentiated at 21 dps, under control conditions, and after DSS exposure. Fluorescent immunocytochemistry demonstrated that Caco-2 untreated monolayers displayed an almost homogeneous intracellular distribution of cytoplasmic SLC15A4/PHT1 localization (Figure 3), and few immunoreactive vesicular formations with high SLC15A4/PHT1 content primarily localized in the perinuclear compartment could be observed (Figure 3 and Figure 4). Remarkably, 2% DSS treatment resulted in cytoplasmic immunoreactivity that remained relatively intact with respect to the untreated monolayer (Figure 3) but clusters of cells displayed highly increased presence of immunoreactive vesicles localized in perinuclear domains, at a much greater extent than untreated monolayer (Figure 3 and Figure 4). After 4% DSS exposure, further enhancement of SLC15A4/PHT1 cytoplasmic immunoreactivity has been observed with respect to control and 2% DSS-treated cells (Figure 3). Immunoreactive vesicles were always abundantly observed but clearly distributed in all the cytoplasm and more intensely labeled compared to the other experimental conditions (Figure 4).

Quantitative analysis of fluorescence intensity was performed and further confirmed that SLC15A4/PHT1 immunoreactivity was significantly increased in 4% DSS-treated monolayers (331.2% ± 15.6) respect to 2% DSS-treated (86.5% ± 5.6; *p* < 0.0001) and control monolayer (100% ± 3.8; *p* < 0.0001) (Figure 4).

### 2.4. SLC15A4/PHT1 Expression in IBD Ulcerative Colitis (UC) Patients

Firstly, the biopsy specimens of UC patients were reviewed for evidence of microscopic features of the UC disease. Figure 5 shows evidence from UC patients which are representative of a group of n = 20. In UC patients, epithelium showed mucosal distortion with surface irregularity and alterations in crypt size, spacing, orientation (i.e., loss of parallelism), and shape. The increased distance and the separation of the crypts from the underlying *Muscolaris mucosae* have been also detected. Moreover, a reduction in the number of Goblet cells can be observed, which appear hyperplastic. Histological studies revealed that, in patients with active UC, SLC15A4/PHT1 immunoreactivity was strongly and specifically localized in the absorptive epithelium (Figure 5A) and distributed in small intracellular vesicles near the brush border of the enterocytes (Figure 5B). Sporadic immunoreactive patches can be observed in immune cells of the lamina propria (Figure 5A,B) and in Goblet cells (Figure 5C).

SLC15A4/PHT1 immunoreactivity was found higher in inflamed colon epithelium of UC patients than in healthy controls (Figure 6). Biopsy specimens of healthy control patients showed a normal mucosa in which the crypts are uniformly spaced, arranged perpendicular to the *Muscularis mucosae,* and the crypt bases are in contact with the upper edge of the *Muscularis mucosae.* Moreover, Goblet cells appear regular in number and morphology. In healthy epithelium, SLC15A4/PHT1 immunoreactivity is very low with respect to UC patients; in fact, it is detected only in limited regions of the mucosal surface, within the enterocyte cytoplasm near the brush border (Figure 6). Quantitatively, the SLC15A4/PHT1 expression/nuclei ratio confirmed that SLC15A4/PHT1 expression is significantly increased in UC patients (0.20 ± 0.03) compared to healthy control (0.08 ± 0.01; *p* < 0.01; see graph in Figure 6).

From the corresponding histological sections of active UC patients and healthy controls included in the study, total RNA was extracted in order to analyze the expression levels of SLC15A4/PHT1 mRNA. qPCR analysis revealed that the expression levels of the gene are significantly upregulated in UC patients compared to control patients (370% ± 150 vs. 100% ± 23; *p* < 0.05; see Figure 7).

## 3. Discussion

SLC15A4/PHT1 (solute carrier family 15 member 4; peptide histidine transporter 1) is an endo-lysosomal carrier involved in TLR7-9/NOD1 signaling pathways in immune cells and intestinal epithelial cells. In humans, it has been investigated with respect to autoimmune syndromes, including IBD. Although in this pathophysiological context some mechanisms and pathways involving SLC15A4/PHT1 in immune cells have been described, knowledge of its gene products in cells of the gastrointestinal tract is still poor. Here, we investigated *SLC15A4/PHT1* gene products in vitro in differentiated enterocyte-like Caco-2 monolayers, as a model of the intestinal epithelial layer, responding to proinflammatory DSS (dextran sodium sulphate). Compromised epithelium/barrier function is a feature of inflammatory intestinal disease and may be critical to both initial pathogenesis and reactivation of chronic intestinal disorders [17]. In this view, DSS is a chemical agent specifically mirroring inflammatory processes of IBD in vivo in mouse models of ulcerative colitis, which has been widely used to induce epithelial barrier disruptive inflammation in vivo and in vitro [18,19], although many mechanisms associated with DSS-induced colitis are still to be clarified.

We challenged differentiated enterocyte-like Caco-2 monolayers in order to elicit a framework of inflammation within which assessing SLC15A4/PHT1 modulation and localization i.e., its (dys) regulation in inflamed vs. healthy epithelial layer. As contemplated in the Introduction, SLC15A4/PHT1 involves TLR7 and MTORC1 which run also on NfKB-independent inflammation routes. In our in vitro intestinal cell model, we analyzed NfKB1, IL-6, and IL-1 mRNAs for checking the “ab initio” responsiveness in terms of inflammation activation. In this framework, NF-kB1 and IL-6mRNA levels significantly increased by all treatments in a DSS dose-dependent manner (see Figure 1), together with CASP3 mRNA, indicating that activations of inflammation and apoptosis pathways concur in Caco-2 monolayers challenged with DSS, as reported by other studies [20,21]. In our in vitro model, mRNA expression variations of the *SLC15A4/PHT1* gene have been detected. The SLC15A4/PHT1 mRNA increases in DSS dose-dependent manner (see Figure 2). These results are, remarkably, in agreement with the immunofluorescence analysis in which anti-human SLC15A4/PHT1 immunoreactivity increases in DSS-treated monolayers depending on dose (Figure 3 and Figure 4). Moreover, SLC15A4/PHT1 immunoreactivity increases in DSS dose-dependent manner feasibly according to DSS-induced morphological derangement of monolayers, although this qualitative association deserves further quantitative/molecular assessment. From the overall data emerges that, in parallel to NF-kB1, IL-6, and CASP3 mRNA expression, DSS treatments coherently increase the amount of the *SLC15A4/PHT1* gene products in enterocyte-like monolayers; so, the in vitro model hints a potentially amplified function of SLC15A4/PHT1 associated to inflammation triggering in enterocyte-like cells.

In parallel, investigations have been moved to the human clinical setting. In light of studies that have linked SLC15A4/PHT1 to inflammatory diseases [10,16] and IBD [6], here it has been investigated for the first time the expression of SLC15A4/PHT1 protein in colon histological sections of IBD patients with ulcerative colitis (UC), finding its major localization in epithelial cells. Then, of great importance was the evidence of subcellular localization of SLC15A4/PHT1. In previous studies, the cellular localization of the transporter was studied by immunofluorescence only in epithelial cell lines demonstrating that this protein was expressed in early endosomes [12]. In human colon biopsies from UC patients, SLC15A4/PHT1 immunoreactivity was localized in absorptive enterocytes (near brush border) putatively in intracellular vesicular formations in agreement with its endo/lysosome localization, and rarely in secretory cells (Goblet cells) besides in immune cells of lamina propria (Figure 5). Histological analysis revealed that anti-human SLC15A4/PHT1 immunoreactivity has been found abundant along the entire inflamed epithelial layer of UC patients with respect to healthy control patients, in which it is only faintly detected (Figure 6). The intense SLC15A4/PHT1 labeling in colon biopsies of UC patients hints at confirmation of the assessment of SLC15A4/PHT1 as a marker of intestinal monolayer’s response to inflammation, as strongly corroborated by the results of in vitro studies presented. It is worth noting that the DSS Caco-2 monolayer model is shown to be valuably predictive, in light of the findings in vivo in human patients. According to histochemistry data, qPCR analysis revealed that the expression of SLC15A4/PHT1 mRNA increases in UC patients with respect to healthy controls (Figure 7); this evidence is in accordance with Lee et al. which preliminarily described the up-regulation of SLC15A4/PHT1 mRNA in inflamed areas of the colon in UC and Chron’s disease patients [12]. Taken together, the mRNA/protein up-regulations are found according to a hypothesis of gain of SLC15A4/PHT1 function associated with GI inflammation pathways and, particularly, to IBD; dysregulation, i.e., increased expression of SLC15A4/PHT1 is revealed as deeply associated to inflammation in patients’ intestinal epithelium, allowing to point *SLC15A4/PHT1* gene as a novel marker to be exploited in studying IBD manifestation.

## 4. Materials and Methods

### 4.1. Reagents and Materials

All chemicals, reagents, and kits were purchased at cell culture/molecular biology grade. Plasticware were invariably purchased sterilized, disposable, and treated for cell culture. Fetal bovine serum (FBS), Dulbecco’s phosphate buffer saline (D-PBS), Eagle’s minimum essential medium (MEM), penicillin/streptomycin solutions, trypsin, L-glutamine, and non-essential amino acids were purchased from Corning-Fisher Scientific (Rodano, MI, Italy). 4’,6-Diamidino-2-Phenylindole (DAPI; Cat.: 28718-90-3), Triton X-100, paraformaldehyde (PFA; Cat. 30525-89-4), bovine serum albumin (BSA), dextran sulfate sodium salt from Leuconostoc spp. (DSS Mr 5000; Cat.: 9011-18-1) were obtained from Sigma-Aldrich (Milano, Italy).

### 4.2. Cell Culture and Treatments

Human epithelial Caco-2 cells (ATCC n. HTB-37™) were grown at 37 °C, in a humidified atmosphere (5% CO_2_ in air), in MEM supplemented with 10% (*v*/*v*) FBS, 2 mM L-glutamine, 100 µg/mL penicillin-streptomycin and 1% (*v*/*v*) non-essential amino acid mix solution. The culture medium was replaced every third day and propagation occurred routinely every 4–5 days post-seeding (dps). For the experimental treatments, cells between passage 3 and passage 10 of propagation were used, after continuous growth for 21 dps in standard culture conditions, in order to obtain a spontaneously differentiated enterocyte-like intestinal epithelial monolayer according to standard Caco-2 cell differentiation protocols [22,23]. At 21 dps, monolayers grown in 12-well plates (Corning-Fisher Scientific, Rodano, MI, Italy; seeding density 0.5 × 10^5^ cells per well) were incubated for 48 h in the presence of 2% or 4% (*w*/*v*) DSS (final concentrations in the culture medium); 100X DSS solutions in D-PBS were filtered through 0.22 µm pore sterile filters before use.

### 4.3. Immunocytochemistry (ICC)

Briefly, 1.5 × 10^5^ Caco-2 cells were seeded and grown on autoclaved, UV-sterilized glass coverslips for 21 dps, then fixed with 4% (*w*/*v*) paraformaldehyde (PFA) after the experimental treatments as described above. Cells were washed with D-PBS (3 times for 10 min), permeabilized with 0.25% (*v*/*v*) saponin in D-PBS for 10 min at room temperature, and blocked against unspecific binding with 5% (*w*/*v*) bovine serum albumin (BSA) and 0.1% (*v*/*v*) Triton X-100 in D-PBS for 30 min at room temperature. Afterwards, the cells were incubated with the anti-SLC15A4 primary antibody (Cat. n. #PA5-42513, ThermoFisher;1:200 dilution) overnight at 4 °C. After washing, the cells were incubated with the anti-rabbit IgG F (ab`)_2_ fragment-FITC secondary antibody (Cat. n. F1262, Sigma-Aldrich; 1:500 dilution) for 1 h at 37 °C. A technical negative control was performed by omitting primary antibody and replacing it with D-PBS. Following ICC protocol, nuclei were counterstained by incubating cells for 5 min with DAPI (0.1 mg/mL in D-PBS, l_ex_ 340 nm, l_em_ 488 nm). Coverslips were finally mounted with 1:1 glycerol/D-PBS on glass slides and examined with an LSM 710 Zeiss confocal laser microscope equipped with the Zen2012 Black Edition program (Zeiss, Dresden, Germany). The quantification of immunofluorescence in acquired pictures was carried out using the ImageJ open-source program (https://imagej.net/ImageJ (accessed on 27 December 2021). In each image, ten digital areas were selected in appropriate regions and corresponding fluorescence intensity was calculated as integrated density (i.e., area value multiplied by mean gray value). For each of the three biological replicates, the average values of fluorescence intensity were expressed as a percentage with respect to the untreated control cells (100%).

### 4.4. Immunohistochemistry (IHC) on Biopsy Sections from IBD Ulcerative Colitis Patients

Ulcerative colitis (UC) patients and healthy controls (HC) were retrospectively enrolled at the Division of Gastroenterology and Digestive Endoscopy of the National Institute of Gastroenterology “S. de Bellis”, Castellana Grotte, Bari, Italy. Written informed consent was obtained from all study participants. The study was carried out according to the principles of the Declaration of Helsinki and was approved by the local Institutional Ethics Review Boards.

Formalin-fixed and paraffin-embedded tissue blocks were obtained from 40 patients divided into two groups:Healthy Control group (n = 20 patients) including intestinal surgical resections from patients without ulcerative colitisUlcerative colitis group (n = 20 patients, mean age 61 years, range 33–80) including total or subtotal proctocolectomies from patients with long-standing ulcerative colitis refractory to medical therapies.

Sections stained with hematoxylin and eosin were reviewed by a pathologist to confirm the adequacy of the sample and to evaluate the morphologic and or pathological characteristics of the samples.

For IHC detection, 4 µm sections were cut and mounted on Apex Bond IHC slides (Leica Biosystems, Buffalo Grove, IL, USA). Tissue sections were incubated with human anti-SLC15A4 polyclonal antibody (Cat n. PA5-42513 Invitrogen, Waltham, MA, USA, 1:100 dilution) for 30 min at room temperature. Antigen retrieval was performed by BOND Epitope Retrieval Solution 2 using EDTA pH 9. The Bond Polymer Refine Detection Kit (Leica Biosystems, Buffalo Grove, IL, USA) was used as a visualization and chromogen reagent according to the manufacturer’s instructions. The samples were recorded as negative when the number of stained cells was less than 5%.

### 4.5. RNA Extraction

RNA extractions from cell cultures were performed using the All-Prep DNA/RNA/Protein mini kit (Qiagen, Hilden, Germany) according to the manufacturer’s instructions. Briefly, cells grown in multi-well plates were washed twice with D-PBS and then lysed with the kit lysis buffer by scraping directly on the plate surface. At the end of the RNA/protein extraction protocols, RNA aliquots were stored in RNase-free conditions at −80 °C until use. RNA concentrations were calculated by spectrophotometry, and the λ_260_/λ_280_ ratios were calculated to evaluate RNA purity; all the RNA extractions were qualitatively tested by electrophoresis of RNA samples on 1% (*w*/*v*) agarose gels. Total RNA extraction from FFPE histological sections of 5 μm thickness was performed with the miRNeasy FFPE Kit (Qiagen, Hilden, Germany) according to the manufacturer’s protocol including the treatment of sections with Deparaffinization Solution (Qiagen, Hilden, Germany). Total RNA was then eluted in ribonuclease-free water. The RNA concentration was determined with the NanoDrop ND-2000 Spectrophotometer (Nanodrop Technologies, Wilmington, DE, USA).

### 4.6. Primer Design, Reverse Transcription and Real Time PCR (qPCR) Assays

The mRNA reference sequences of the investigated genes were collected from the GenBank database (https://www.ncbi.nlm.nih.gov/gene) and were used to select oligonucleotide sequences as primer pairs for consequent real-time PCR (qPCR) assays. By mRNA-to-genomic sequence alignment, the gene-specific forward and reverse primers were designed on different exons (intron spanning) to avoid amplification of genomic DNA. The AmplifX software version 2.0.7 (free download at https://inp.univ-amu.fr/en/amplifx-manage-test-and-design-your-primers-for-pcr) was used to test PCR size, GC content, end stability, self/cross-dimer formation, and melting temperature for the selected primer pairs. Details of the gene-specific oligonucleotide sequences are reported in Table 1. Reverse transcription on the extracted RNAs was performed on 500 ng total RNA for each sample, using the iScript Select cDNA Synthesis Kit (Bio-Rad, Segrate, MI, Italy) according to the manufacturer’s instructions, with random primers in the reaction mix. Before qPCR analysis, primer pairs were tested for efficiency, according to the amplification efficiency parameters for genes of interest and internal controls proposed by Schmittgen and Livak [19]. qPCR assays were performed using the iTaq Universal SYBR Green Supermix (Bio-Rad) with the CFX96 Touch™ Real-Time PCR Detection System (Bio-Rad). In the qPCR analysis, gene expression relative quantification was performed by analyzing the threshold values (C_T_) with the comparative C_T_ method (also referred to as the 2^−ΔCT^ or 2^−ΔΔCT^ method), and qPCR data shown were the 2^−ΔCT^ values, which are considered as proportional to the amount of detected target mRNA. For each target gene and internal control (housekeeping), ΔC_T_ values (ΔC_T_ = target gene C_T_ − housekeeping gene C_T_) were obtained from 2 different rounds of qPCR for each of the three biological replicates. Statistical analysis was performed after the 2^−ΔCT^ transformation [24].

### 4.7. Statistical Analysis

Unless otherwise stated, all data were expressed as the means ± standard error of the mean (S.E.M.). Data means derive from two independent assays for each of the three biological replicates. Statistical analysis by two-tailed unpaired Student’s *t*-test or one-way ANOVA followed by Dunnett’s multiple comparison test was performed using GraphPad Prism 9.4.0; *p*-value ≤ 0.05 were considered significantly different.

## 5. Conclusions

Based on the experimental evidence, an inflammatory context in which the increase of SLC15A4/PHT1 protein product is favored, suggesting a “gain of function” associated with inflammatory and/or autoimmune onset (already hypothesized for SLC15A4/PHT1 in immune cells) has been here putatively evidenced in intestinal cells. The identification and localization of SLC15A4/PHT1 protein were established in colon biopsy specimens from IBD patients with UC and control healthy patients. In UC patients, SLC15A4/PHT1 protein gains expression reaching highly increased levels. Taken together, the research results demonstrate that dysregulation (i.e., increase) of *SLC15A4/PHT1* gene products in the human intestinal epithelial layer is in the framework of GI inflammation, and, particularly, of IBD, leading the way for the assessment of SLC15A4/PHT1 as a marker of inflammation for UC patients.

## Figures and Tables

**Figure 1 ijms-23-13170-f001:**
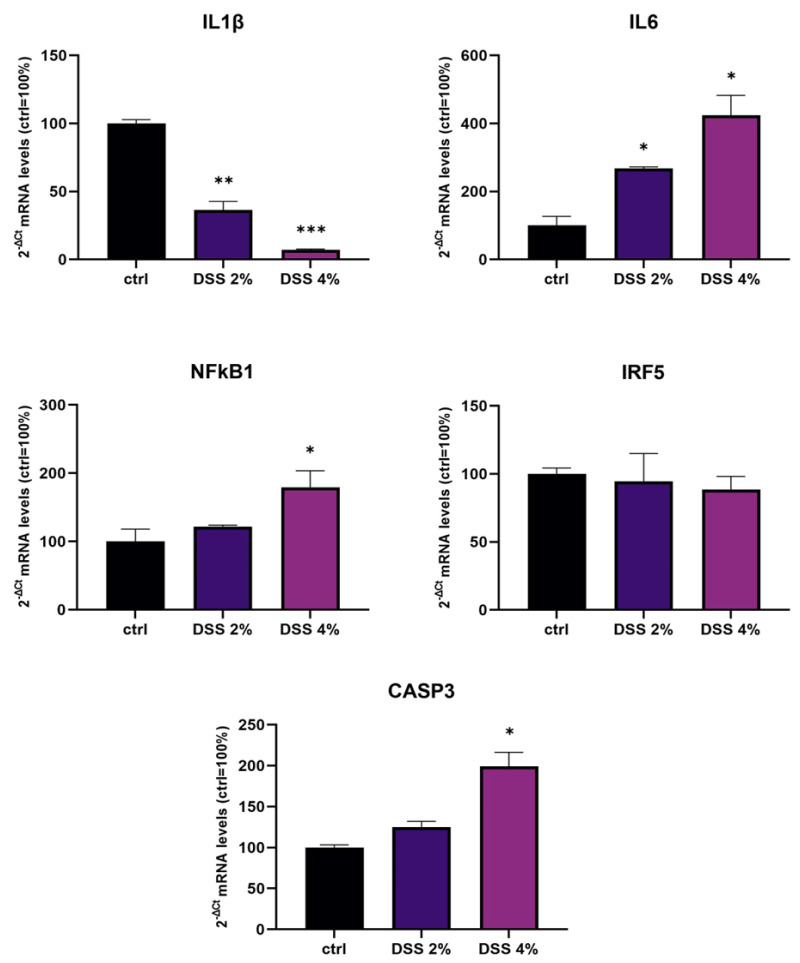
mRNA expression analysis of inflammation and apoptosis-related genes in differentiated Caco-2 monolayers after DSS treatments. mRNA expression analysis by qPCR on total RNA extracted from Caco-2 cell monolayers treated with DSS 2% and 4% for 48 h. Amounts of target IL-1β, IL-6, NF-kB1, IRF5, and CASP3 mRNAs are expressed as 2^−ΔCt^ mean values obtained from 2 rounds of real-time PCR assays for each of 3 independent biological replicates (see Materials and Methods for details) and normalized with respect to the *GAPDH* gene (housekeeping), then expressed as percent with respect to the untreated control (ctrl = 100%). Statistical analysis by one-way ANOVA with Dunnett correction for multiple comparisons (* *p* < 0.05; ** *p* < 0.01; *** *p* < 0.001).

**Figure 2 ijms-23-13170-f002:**
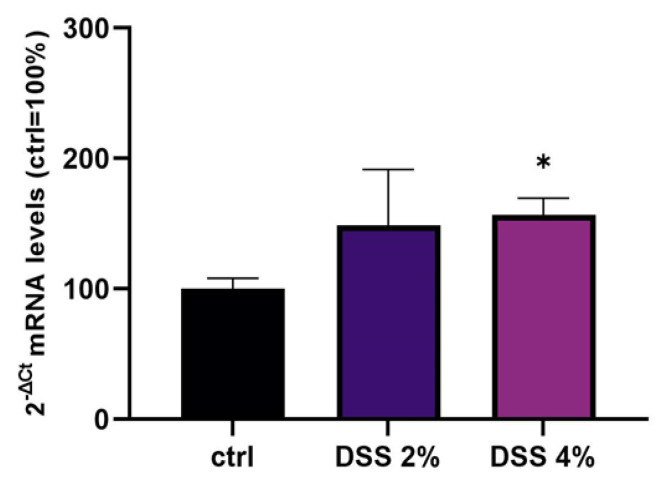
Expression analysis of SCL15A4/PHT1 mRNA in differentiated Caco-2 monolayers after DSS treatments. Expression analysis by qPCR of SCL15A4 mRNA in Caco-2 cell monolayers treated with DSS 2% and 4% for 48 h. Amounts of target mRNA are expressed as 2^−ΔCt^ mean values obtained from 2 rounds of real-time PCR assays for each of 3 independent biological replicates (see Materials and Methods for details) and normalized with respect to the *GAPDH* gene (housekeeping), then expressed as percent with respect to the untreated control (ctrl = 100%). Statistical analysis by one-way ANOVA with Dunnett correction for multiple comparisons (* *p* < 0.05).

**Figure 3 ijms-23-13170-f003:**
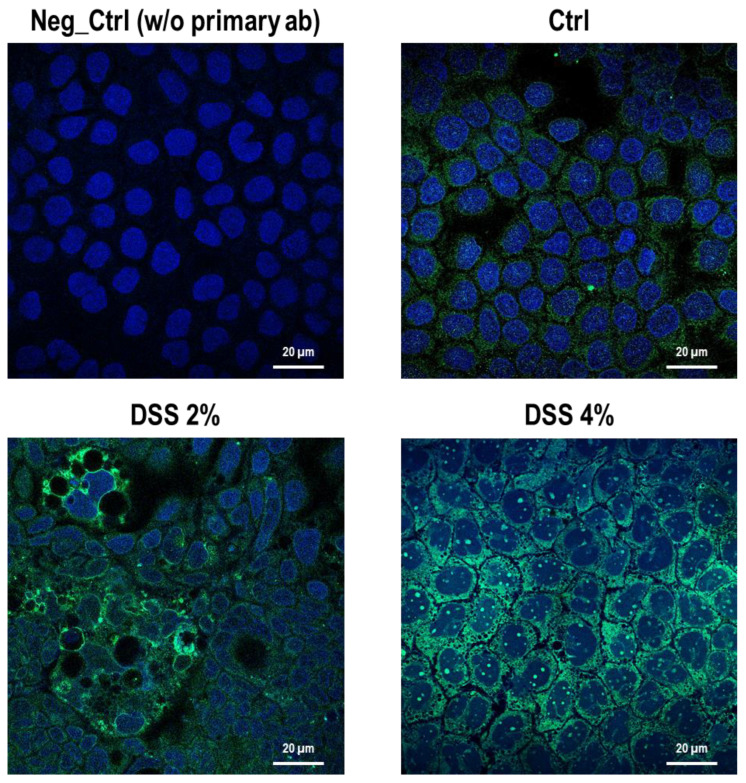
Distribution and localization of anti-human SLC15A4/PHT1 immunoreactivity in enterocyte-like Caco-2 monolayers treated with DSS for 48h. Representative pictures of SCL15A4/PHT1 immunoreactive Caco-2 cells in absence (Ctrl) or presence of 2% or 4% DSS in culture medium. Negative control (Neg Ctrl) represents cell samples incubated in the absence of the primary antibody. Goat anti-rabbit FITC conjugated and DAPI were used for fluorescence detection of cells (green) and nuclei (blue). Scale bar: 20 µm. Magnification: 63X (oil immersion). Images examined with LSM 710 Zeiss confocal laser microscope equipped with the Zen2012 Black Edition program (Zeiss, Dresden, Germany).

**Figure 4 ijms-23-13170-f004:**
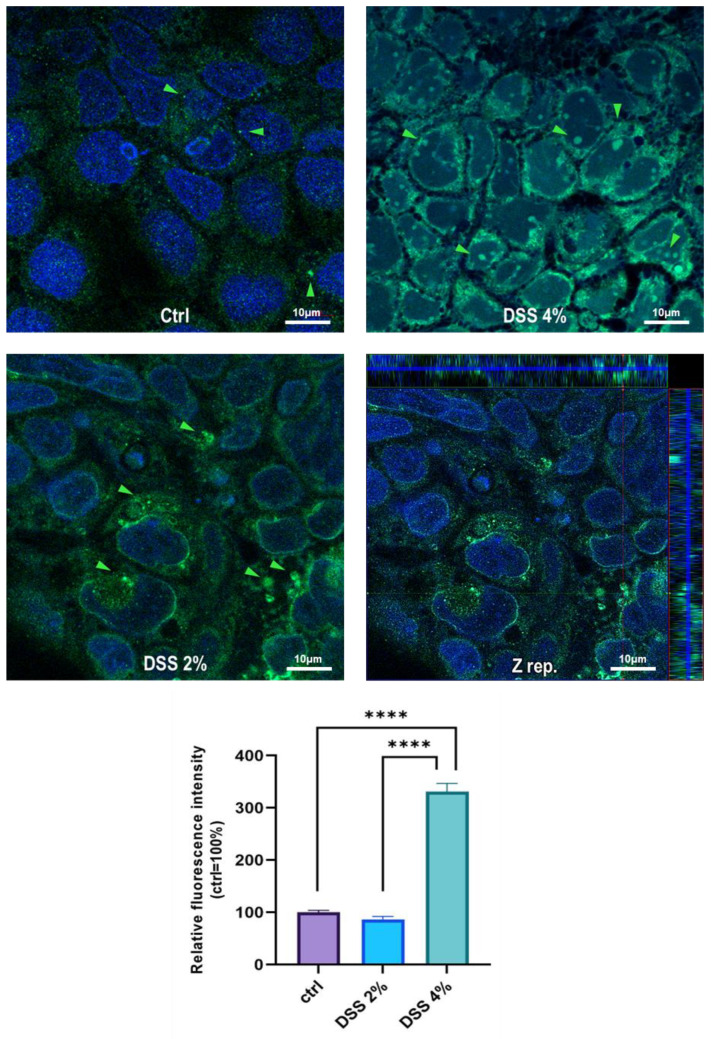
Distribution and localization of anti-human SLC15A4/PHT1 immunoreactivity in enterocyte-like Caco-2 monolayers treated with DSS for 48 h. In untreated cells (Ctrl), the SLC15A4 immunoreactivity is distributed homogeneously within the cytoplasm. After DSS administration, immunoreactivity shows different patterns in treated monolayers depending on dose: variable size vesicles of SLC15A4/PHT1 positive inclusions become more dominant according to DSS concentration. Z-stack representation (Z rep.) demonstrates the intracellular localization of immunoreactivity. Scale bar: 10 µm. Magnification: 63X oil immersion. In histogram: quantitative analysis of SLC15A4/PHT1 fluorescence intensity (see Materials and Methods for detail). Data are presented as mean ± SEM of 5 measures from each of 3 independent biological replicates. Mean values are reported as percent *vs.* untreated control (100%). Statistical analysis: one-way ANOVA with Dunnett correction for multiple comparisons (**** *p* < 0.0001).

**Figure 5 ijms-23-13170-f005:**
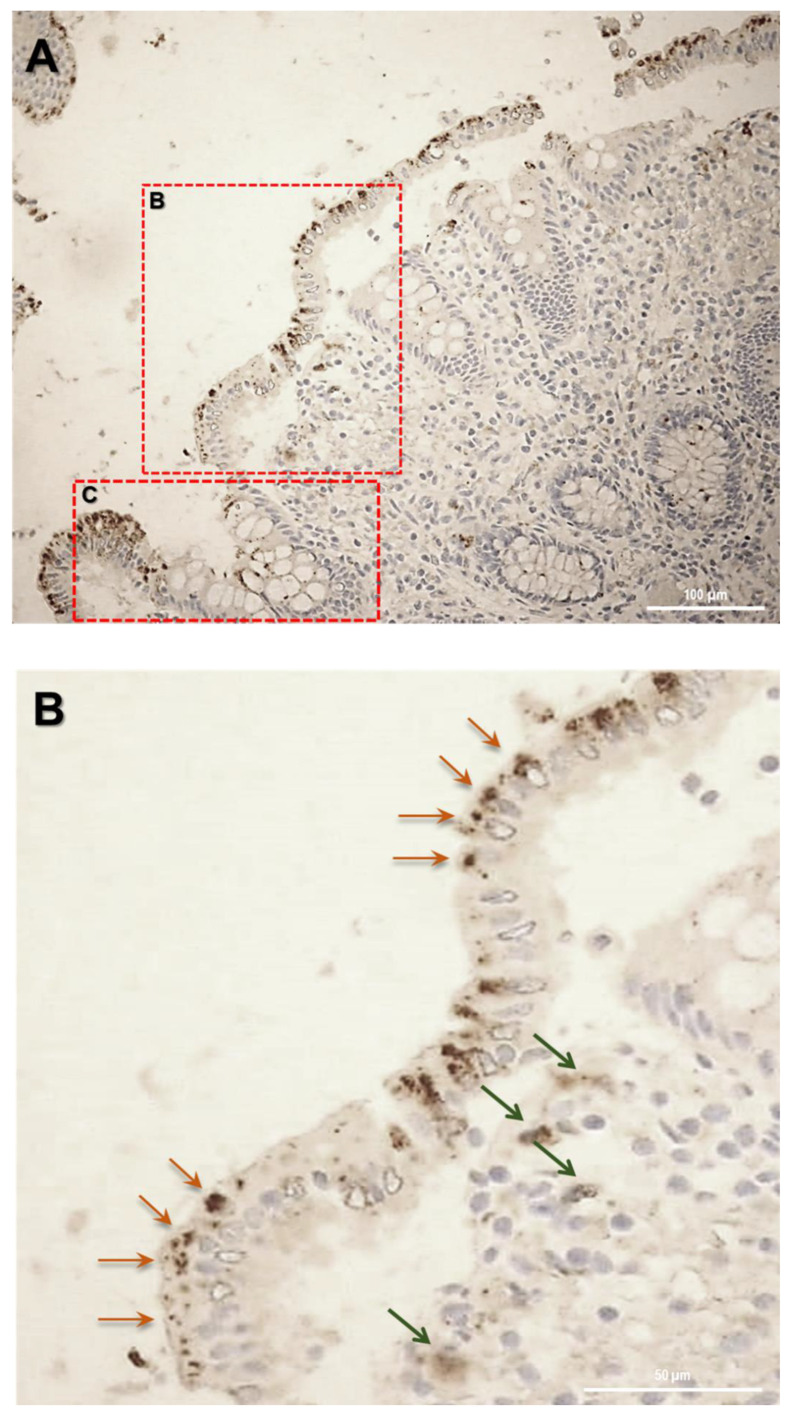
Study of the SLC15A4/PHT1 protein in human colon biopsies from UC patients. (**A**) Localization of SLC15A4/PHT1 by immunohistochemistry in the colon of UC patients. It is possible to note architectural alterations of colonic mucosa, including shortening of crypts, variations in the size and shapes of crypts, and Goblet cell hyperplasia. Regions in squares have been magnified in the pictures below; (**B**) SLC15A4/PHT1 immunoreactivity is localized in small intracellular vesicles near enterocytes brush border, along the mucosal surface (red arrows). Positive inclusions are rarely observed in immune cells of the lamina propria (green arrows); (**C**) positive intracellular formations are localized also in Goblet cells; (**D**) histological section of a UC patient with severe alteration of colonic mucosa. Epithelium surface displays a considerably high SLC15A4/PHT1 expression in vesicles near enterocytes brush border. Scale bar: 100 µm (**A**); 50 µm (**B**–**D**).

**Figure 6 ijms-23-13170-f006:**
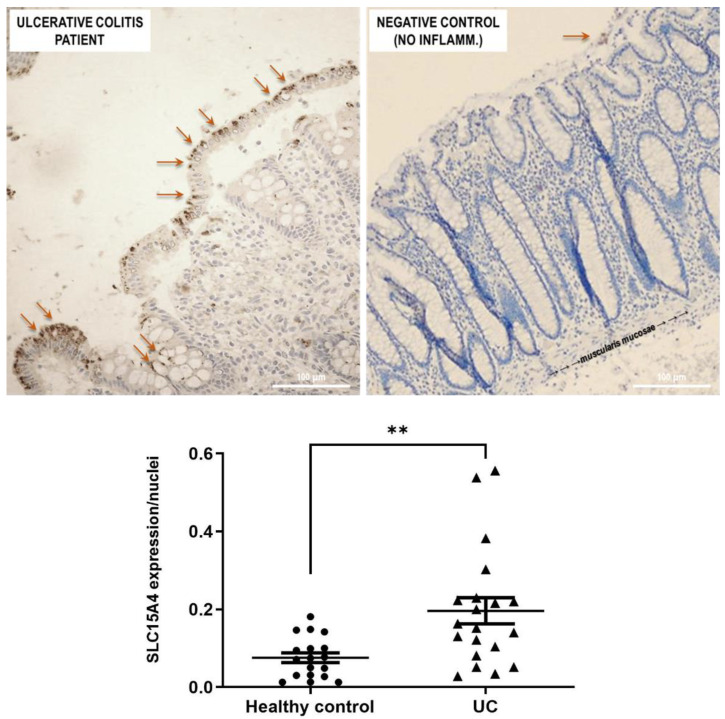
Study of the SLC15A4/PHT1 protein in human colon biopsies from UC patients and healthy controls. Detection of SLC15A4/PHT1 by IHC is strongly reduced in healthy controls (negative control) with respect to UC patients. Immunoreactivity is localized in small intracellular vesicles near enterocytes brush border, along the mucosal surface (red arrows). In the graph below: quantitative analysis of SCL15A4/PHT1 levels in colon of active UC patients and healthy control patients (i.e., without intestinal inflammation). Quantification of staining intensity per nuclei count was performed with ImageJ software. Data are expressed as mean ± SEM (n = 20 for UC and Healthy control). Statistical analysis by two-tailed unpaired Student’s *t*-test (** *p* < 0.01).

**Figure 7 ijms-23-13170-f007:**
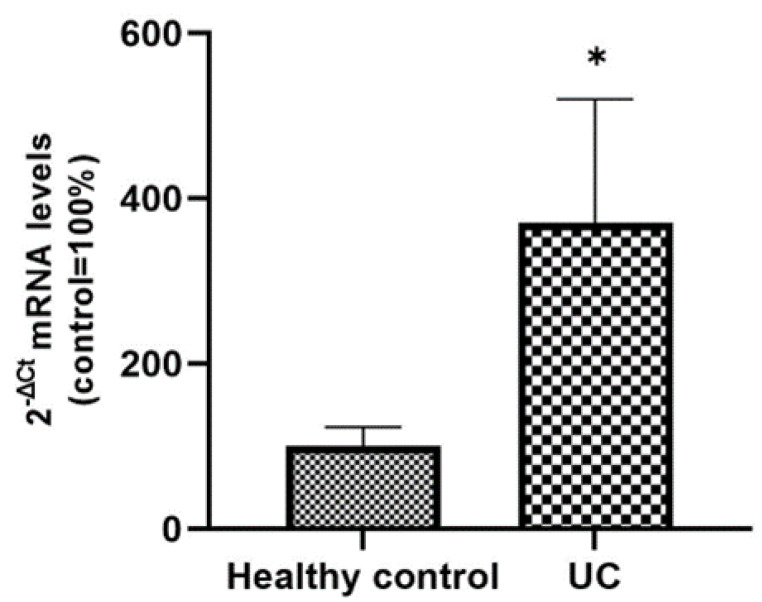
Expression analysis of SCL15A4/PHT1 mRNA in patients with UC and healthy controls. mRNA expression analysis by qPCR on total RNA extracted from histological sections of UC and control patients. Amounts of target mRNA were calculated as 2^−ΔCt^ mean values obtained from 2 rounds of real-time PCR assays for each of the patients enrolled (see Materials and Methods for details) and normalized with respect to the 18S rRNA gene (housekeeping). Values were expressed as percent with respect to the healthy control patients (ctrl = 100%). Statistical analysis by two-tailed unpaired Student’s *t*-test (* *p* < 0.05).

**Table 1 ijms-23-13170-t001:** Features of primer sequences for qPCR expression analysis. For each gene, the NCBI accession numbers of the mRNA reference sequence (RefSeq mRNA) used for primer design are reported. For each primer, the 5’-3’ nucleotide sequence and melting temperature (Tm) are reported. For each mRNA detection, the expected amplicon length is reported (PCR size) in base pairs (bp).

**GENE**	RefSeq mRNA Acc. No.	Sense Primer 5’-3’ (Tm)	Antisense Primer 5’-3’ (Tm)	PCR Size (bp)
*NFKB1b*	NM_003998.4	AATGCCTTCCGGCTGAGTC (59°)	AGGCTGCCTGGATCACTTCA (60°)	140
*IL1b*	NM_000576.3	CCTTCATCTTTGAAGAAGAACC (51°)	GAGGTGGAGAGCTTTCAG (51°)	158
*IL6*	NM_000600.5	GATGCTTCCAATCTGGATTC (51°)	CAGGAACTGGATCAGGAC (51°)	164
*IRF5*	NM_001347928.2	CTCAATGAGCTCATCCTGTTC (53°)	CTGAGAACATCTCCAGCAGC (55°)	169
*CASP3*	NM_001354777.2	CTGGACTGTGGCATTGAGAC (59 °C)	CAAAGCGACTGGATGAACC (57 °C)	157
*SLC15A4/PHT1*	NM_145648.4	TGAAGGCATTGGAGTCTTT (51°)	TGGAAATACACTGTCCAGTAA (51°)	168
*GAPDH*	NM_002046.7	AAACCTGCCAAGTATGATGA (51 °C)	TACTCCTTGGAGGCCATGT (54 °C)	217

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
