# Peer review of "Identification of SLC15A4/PHT1 Gene Products Upregulation Marking the Intestinal Epithelial Monolayer of Ulcerative Colitis Patients"

_ijms, 2022, doi:10.3390/ijms232113170_

Round 1
Reviewer 1 Report
Fig-1. After DSS treatment of CaCo2 cells, dose dependent increase in IL-6 and NFKb mRNA expression were observed. However, IL-1 beta mRNA expression showed a dose dependent decrease pattern. Could you explain?
Author Response
We thank the reviewer for giving us the possibility to further explain these specific results. Indeed, our data on IL-1 beta mRNA expression in Caco-2 cells are quite interesting and deserve to be further elucidated, potentially representing an aspect for clarifying “immune cell-like”pathways and behaviours of enterocyte-like cells. In fact, we will go through this issue in the research actions to follow up this manuscript, although in this work we did not deepen information on the specific IL-1 beta mRNA trend. Nevertheless, it is worth to note that in literature data on the expression of IL-1 beta mRNA in inflammation-triggered Caco-2 cells can vary depending on the culture conditions of cell lines.
E.g., increased expression of IL-1 beta mRNA is reported in Caco-2 cells simply utilized at confluence without reaching spontaneous enterocyte-like differentiation at longer times [see e.g. Zhao Q et al. Inflamm Res. 2018 Aug;67(8):663-670. doi: 10.1007/s00011-018-1155-6. Epub 2018 May 15. Retraction in: Inflamm Res. 2022 Jun;71(5-6):727. PMID: 29766204; PMCID: PMC6028846; Zhao Het al. Int Immunopharmacol. 2016 Oct;39:121-127. doi: 10.1016/j.intimp.2016.07.020. Epub 2016 Jul 26. PMID: 27472293], whilst we do work on monolayers at 21 days post seeding (dps), i.e., the stage of enterocyte-like spontaneous differentiation of the Caco-2 cell type: this simple difference might imply differential activation and intertwinement of intracellular pathways in intestinal epithelial cells.
Moreover, in literature other differential regulations of cytokines (including IL-1 beta) at mRNA level may arise from the molecular specificity of the (pro)inflammatory agents: for instance, in our case the molecular weight of the DSS utilized (5000 KDa, the more usual as suitable for cell culture) might also be a cause of differential activation of some genes with respect to other DSS molecular weights [see e.g. Chang KW and Kuo CY. Food Funct. 2015 Oct;6(10):3334-41. doi: 10.1039/c5fo00513b. PMID: 26263169; Toutounji M et al. Int J Mol Sci. 2020 Apr 15;21(8):2726. doi: 10.3390/ijms21082726. PMID: 32326391; PMCID: PMC7215722; Roselli M et al. Front Nutr. 2022 Apr 14;9:862974. doi: 10.3389/fnut.2022.862974. PMID: 35495925; PMCID: PMC9047546].
Overall, differentiated Caco-2 monolayers (≥ 21 dps) draw their own immune/inflammation activation scheme (which, anyway, doesn’t comprise the same full framework of activated genes as prototypically do immune cells). Remarkably, we invariably found reduction of IL-1 beta mRNA in proinflammation-challenged Caco-2 (21 dps mature) monolayers not only in the research proposed in this manuscript, but also in other experimental settings. In our hands, it seems that IL-1 beta involvement is “under-demanded”; this evidence (together with literature cited above) conversely hints that it a) might be increasingly required in cells cultured for periods shorter than 21 dps, i.e., in cells not reaching the enterocyte-like spontaneous maturation, or b) depends on different molecular features of the stimuli which imply modular intracellular switching of immune/inflammation pathways not always involving all expected partner cytokines and, we must say, not all gene regulations at the transcriptional level.
Besides this, our data hint down-regulation of IL-1 beta mRNA which might be interpreted as a switch-off need of IL-1 beta downstream actions; this does not exclude that the already synthesized cytokine is massively released to act (indeed, many research article show DSS-induced IL-1 beta RELEASE without associating data on mRNA/protein levels) and this is one of the things we intend to demonstrate in the continuation of the studies.
Reviewer 2 Report
The “Identification of SLC15A4/PHT1 gene products upregulation marking the intestinal epithelial monolayer of ulcerative colitis patients” research article is the first identification of SLC15A4/PHT1 gene in human ulcerative colitis (UC) patients. However, some modifications need to be made and supplements need to be added as follows:
1. In figure 5, quantification of SLC15A4/PHT1 staining intensity per nuclei count in human colon biopsies from UC patients and healthy controls. This figure cannot improve the protein expression. And All figures showing mRNA data should show protein SLC15A4/PHT1 protein expression using Western blot.
2. Recently other published papers have shown that SLC15A4 inflammasome function via mTORC1 signaling in dendritic cells (Cynthia et al., BioRxiV. 2022), and IL-6 and IL-1b were not regulated inflammation via SLC15A4 in SLC15A4feeble mice compared to C57 mice (Alexis et al., Scientific reports. 2018). However, the figure only showed the expression of anti-inflammation cytokine (IL-1B) and pro-inflammation cytokine (IL-6, NF-kB), and Caspase-3 (apoptosis). What is the SLC15A4 pathway in the regulation of inflammation? Need to discuss the SLC15A4 pathway in inflammation.
3. Need the measurements of cell death or cell viability after being treated with DSS. The figure 1 to 4, Caco-2 cells were treated with DSS 2% and 4% for 48 hr. However, this article did not show the cell viability of DSS-treated cells.
4. Author should prove the human biopsies from UC patients and healthy controls used for this study by providing the expression of IL-6, IL-1B, and NF-kB.
Author Response
- In figure 5, quantification of SLC15A4/PHT1 staining intensity per nuclei count in human colon biopsies from UC patients and healthy controls. This figure cannot improve the protein expression. And All figures showing mRNA data should show protein SLC15A4/PHT1 protein expression using Western blot.
We thank the reviewer for giving us the possibility to better specify this point. In Figure 5, we have shown representative images of the immunolocalization of SLC15A4/PHT1 in human colon biopsies from UC patients and healthy controls. The quantitation by mean intensity of staining per nuclei count in the enterocyte layer of all UC patients and controls was in Figure 6. As in other similar research articles, we decided to evaluate SLC15A4/PHT1 protein expression by IHC since the main aim of this paper was to firstly demonstrate the epithelial localization of this marker in human UC biopsies, and, possibly, differences in protein presence induced by the disease state with respect to controls. Future studies are planned and will be conducted to finely investigate the protein expression levels in different phases of the disease (and/or therapy) also by using Western blot analyses which, on the other hand, have not been contemplated in this work also due to limited quantity of histological material which did not allow simultaneous WB assays. As the same, in our in vitro experiments in DSS-treated Caco-2 monolayers, the immunoreactivity showed by immunofluorescence mainly aimed at describing changes of localization and pattern; nevertheless, under 4% DSS treatment the detected (statistically significant) increase of fluorescence intensity was clearly associated to higher presence of antibody reaction against SLC15A4 protein inside cells, as assessed by the assays with the negative control. Also, in the cell-based experimental set, WB assays were performed but the antibody showed to be not suitable for the WB technique, whilst invariably working very well in IHC/ICC.
- Recently other published papers have shown that SLC15A4 inflammasome function via mTORC1 signaling in dendritic cells (Cynthia et al., BioRxiV. 2022), and IL-6 and IL-1b were not regulated inflammation via SLC15A4 in SLC15A4 feeble mice compared to C57 mice (Alexis et al., Scientific reports. 2018). However, the figure only showed the expression of anti-inflammation cytokine (IL-1B) and pro-inflammation cytokine (IL-6, NF-kB), and Caspase-3 (apoptosis). What is the SLC15A4 pathway in the regulation of inflammation? Need to discuss the SLC15A4 pathway in inflammation.
We thank the reviewer for the opportunity to give some specifications. In the research papers cited by the reviewer (Cynthia López-Haber et al., 2022; Alexis D. Griffith et al., 2018), as well as in other important research articles dealing with SLC15A4 involvement in inflammation/immune/autoimmune pathways, data mainly come from immune cells and specifically dendritic cells. Our work is one of the few first detecting SLC15A4 regulation or dysregulation in intestinal epithelial cells, thus represents a start-up set of data for further elucidating the regulation framework and network including SLC15A4 modular responsiveness in intestinal districts, tissues and cells.
In our Introduction section, we reported and discussed the major evidence as present in literature in research works describing the SLC15A4 involvement in intracellular pathways regulating inflammation in immune cells. To summarize, SLC15A4 regulates both TLR7 activity and mTORC1 activity by modulating the luminal environment of endolysosomes such as pH through amino acid or peptide transportation. The NF-κB pathway and IRF7 pathway are activated downstream of TLR7 and induce pro-inflammatory cytokines and IFN-I, respectively. IFN-I starts the feed-forward circuit through IFNAR–STAT1–IRF7 axis to produce larger amounts of IFN-I. Upon induction of IRF7 by IFNAR signaling, translation of IRF7 proteins is tightly depending on mTORC1 activity, which is further controlled by SLC15A4. SLC15A4 is required for both TLR proximal signaling events and the IFN-I circuit. SLC15A4 involves TLR7 and MTORC1 which are NfKB-independent. Deletion of SLC15A4 or TASL (“TLR adaptor interacting with SLC15A4 on the lysosome”) specifically impairs the activation of the IRF pathway without affecting NF-κB and MAPK signalling, which indicates that ligand recognition and TLR engagement in the endolysosome occur normally (Heinz LXet al. Nature. 2020 May;581(7808):316-322. doi: 10.1038/s41586-020-2282-0. Epub 2020 May 13. PMID: 32433612; PMCID: PMC7610944). Here, in the intestinal cell model in vitro, we analyzed NfKB1, IL-6 and IL-1 mRNAs for checking the “ab initio” responsiveness in terms of inflammation activation; we have made this consideration explicit in the Discussion (see revised lines 265-271 in track-changes). Based on these premises, we will further deepen the molecular pathways and functional intertwinement for assessing SLC15A4-dependent routes in intestinal epithelial cells, still neither described nor verified in any research present in literature.
- Need the measurements of cell death or cell viability after being treated with DSS. The figure 1 to 4, Caco-2 cells were treated with DSS 2% and 4% for 48 hr. However, this article did not show the cell viability of DSS-treated cells.
We thank the reviewer for giving us the chance to add information. prior to carrying out the experiments, we had already obtained data on cell viability under DSS exposure. We have previously performed MTT assays on DSS-treated Caco-2 monolayers, and a file with those data will be made available to Editor and Reviewers through the submission system. We decided not to insert the MTT assay in the manuscript just because the available literature on the toxicity of DSS in vitro is very extensive, even in our cell model, with all concentrations, all treatment times, all molecular weights of DSS, and for this reason we considered redundant a further information from us on the viability of Caco-2 exposed to DSS.
- Author should prove the human biopsies from UC patients and healthy controls used for this study by providing the expression of IL-6, IL-1B, and NF-kB.
We thank the reviewer for this comment giving us the opportunity to specify the information. In this work, in human samples we did not test for IL-6, IL-1B, and NF-kB expression since the staining for IL-6, IL-1B, and NF-kB requires several weeks. However, as reported in Materials and Methods section, the inflammatory state of patients has been clearly checked and clinically parameterized at the time of diagnosis. The human biopsies from UC patients and healthy controls were stained with hematoxylin and eosin and reviewed by pathologists at the De Bellis Gastroenterology Institute to evaluate the morphological and the pathological characteristics of the samples. In addition, our clinicians have followed up patients with UC for clinical outcomes and, to date, they present the main clinical signs of overt UC.